# An Overview of SARS-CoV-2 and Technologies for Detection and Ongoing Treatments: A Human Safety Initiative

Ramsingh Kurrey * and Anushree Saha *

School of Studies in Chemistry, Pt. Ravishankar Shukla University, Raipur 492010, Chhattisgarh, India
* Correspondence: ramsinghkurrey@gmail.com (R.K.); anu.saha011@gmail.com (A.S.);
  Tel.: +91-8889629675 (R.K.)

**Abstract:** A new class of coronavirus, known as the severe acute respiratory syndrome coronavirus-2 (SARS-CoV-2) has been discovered, which is responsible for the occurrence of the disease, COVID-19. A comparative study with SARS, MERS and other human viruses was conductedand concluded that SARS-CoV-2 spread more rapidly due to increased globalization and adaptation of the virus in every environment. According to recent WHO reports, by 16 May 2021, the current outbreak of COVID-19 had affected over 174,054,314 people and killed more than 3,744,116 people in more than 222 countries acrossthe world. Finding a solution against the deadly COVID-19 has become an enormous challenge for researchers and virologists. A ring vaccination trial, which recruits subjects connected to a known case either socially or geographically, is a solution to evaluate vaccine efficacy and control the spread of the disease simultaneously, although its implementation is challenging. This review aims to summarize the noteworthy features of the world-intimidating SARS-CoV-2 global pandemic along with its evaluation, problems and challenges in the treatment strategies, clinical efficiency and detection methods proposed so far. This paper describes the impact of the lockdown in response to the COVID-19 pandemic on social, economic, health, and National Health Programs in India; possible ways to control the disease are also discussed.

**Keywords:** SARS-CoV-2; COVID-19; problem and challenges; treatment strategies; detection method; worldwide scenario





## 1. Introduction

Coronavirus disease is an infectious disease caused by a newly discovered human coronavirus or Severe Acute Respiratory Syndrome-2 (SARS-CoV-2). Its official name was announced by the World Health Organization (WHO) as "CoronavirusDisease-19" (COVID-19). Since the beginning of the 21st century, three coronaviruses have caused disastrous outbreaks of pneumonia in human beings: Severe Acute Respiratory Syndrome coronavirus (SARS-CoV) and Middle East Respiratory Syndrome coronavirus (MERS-CoV) in March 2002 and 2012, respectively [1]. SARS-CoV-2 are basically enveloped RNA viruses with the largest known RNA genomes (30–32 kb) distributed widely among humans and other mammals, and birds, causing respiratory, enteric, hepatic and neurologic diseases [2–6]. SARS-CoV-2 is predominantly concomitant with upper respiratory tract illnesses ranging from mild to moderate, including common cough and cold. Most of the people may be infected with one or more of these viruses at some point in their lifetime [7]. SARS-CoV-2, having a total of 39 species under the broad realm of Riboviria, belongs to the family Coronaviridae, suborder Cornidovirineae and order Nidovirales reported by Gorbalenya et al. [8]. The virus depicts crown-like spikes on its outer surface and so, it was named as "coronavirus". These viruses are minute in size (diameter 65–125 nm) and contain a single-stranded RNA as their nucleic material (size ranging from 26 to 32 kbs in length) [9,10]. They contain four major structural proteins, namely, the spike (S) protein (that mediates attachment to the host receptor and subsequent fusion of the virus and cell membrane; membrane (M) protein; nucleocapsid (N) protein; and envelope (E)

protein (Figure 1). Recently, the International Committee on Taxonomy of Viruses divided coronaviruses into three genera (alpha, beta and gamma coronaviruses), which correspond to groups 1, 2, 3, within the subfamily coronavirinae, family of coronaviridae, and order (or superfamily) of nidovirales [9–11].

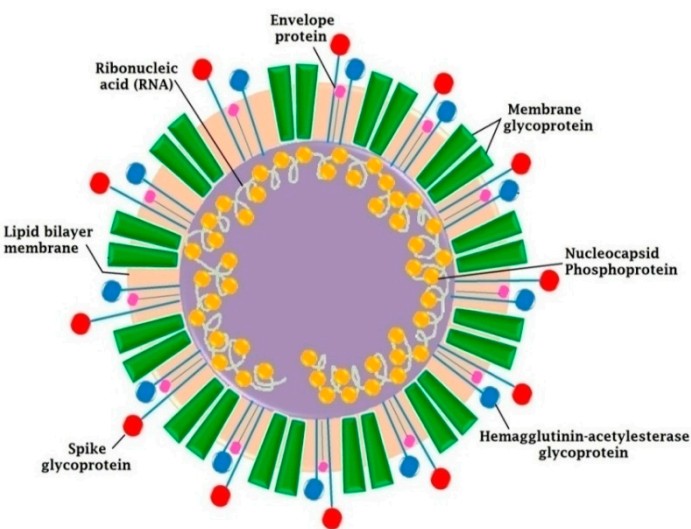

**Figure 1.** A crown-like structural illustration of the novel SARS-CoV-2.

According to WHO, COVID-19 is now a pandemic affecting many countries globally. The harshness of COVID-19 symptoms can range from very mild to moderate respiratory illness and an infected person may recover without requiring special treatment. Some people may have only a few symptoms, while others may have no symptoms at all. The signs and symptoms of COVID-19 might appear 2 to 14 days after exposure. It is noteworthy that coronavirus infections have a range of symptoms, including shortness of breath or difficulty breathing, cough and fever, while other symptoms include headache, loss of smell, sore throat, loss of taste, tiredness, aches, chills, diarrhea and vomiting [9]. The first ever human CoV (HCoV) was recognized in the mid-1960s in human embryonic tracheal organ cultures and until 2003, only two—HCoV-229E and HCoV-OC43—were recognized. Though the earliest human CoVs identified were back in the year 1960 from respiratory infections in adults and children, the chief scientific interest in CoVs research arose only after the appearance of SARS-CoV in 2002–2003 [12]. In this SARS-CoV epidemic, about 8000 human cases were reported along with 774 deaths (around 9.5% mortality rate) [13]. It is notable that in the beginning, SARS-CoV was detected in caged Himalayan palm civets, which were believed to be the natural host of this human pandemic [14]. Subsequent to the SARS-CoV incidence (in 2003), analogous CoVs named HKU3-1 to HKU3-3 were identified in Hong Kong, in horseshoe bats (non-caged) in 2005 [15]. In view of this fact, bats are believed to be the natural host and potential reservoir species which could be considered responsible for any future CoVs epidemics or pandemics [16]. Following the 2003 and 2005 SARS-CoV epidemics, a similar virus in the Middle East region of the world evolved, leading to a brutal respiratory illness which was identified as the MERS-CoV [17]. The mortality in this case was higher than the preceding SARS-CoV pandemic as it accounted for around 919 lives out of the sum total of the 2521 human cases (i.e., about 35% mortality). Presently, seven diverse CoV strains have been identified that are believed to infect humans; these include HCoV-229E, HCoV-NL63, HCoV-OC43 and HCoV-HKU1. However, these strains usually cause self-resolving infections, while in contrast, there are also certain severe acute coronaviruses which can cause lethal respiratory infections in humans, such as SARS-CoV, MERS-CoV and the recently identified SARS-CoV-2 [18]. Subsequent to the discovery of tobacco mosaic virus (TMV) in 1892 and the foot-and-mouth disease virus in 1898, the first 'filterable agent' to be discovered in humans in 1901 was the yellow fever virus [19]. At a rate of three or four per year, the identification of latest species of human virus is still an

ongoing process, and it is worth mentioning that the viruses make up over two-thirds of all new human pathogens [20]. An extremely significant fact is that the majority of human pathogen species are bacteria, fungi or helminths.

The newly discovered SARS-CoV-2 differs highly in importance. Viruses range from the rare and mild illness due to the Menangle virus to the devastating public health impact of COVID-19 as human disease [21]. The recent discovery proves that a novel coronavirus is the probable cause of the newly recognized SARS. Although it is reported that the human coronaviruses cause up to 30 percent of colds and hardly ever cause lower respiratory tract disease. On the other hand, coronaviruses cause devastating epizootics of respiratory or enteric disease in livestock and poultry [22]. Therefore, the global coronavirus disease pandemic (COVID-19) encountered is a very serious concern worldwide. Various review papers taking different approaches were published to provide an overview of coronaviruses. To the best of our knowledge, no systematic reviews have been found in the last few years on SARS-CoV-2 with global health and challenges.

This review aims to summarize the noteworthy features of the world-intimidating SARS-CoV-2 global pandemic along with its evaluation, problems and challenges, treatment strategies, clinical efficiency and detection methods. In addition, the review also includes detailed discussions of the transmission methods along with the recommended precautions, including comparisons of the novel SARS-CoV-2 with other types of HCoVs. In this brief review, we also discuss the impact of the lockdown in response to the COVID-19 pandemic on social, economic, health, and National Health Programs in India. The current gaps in our understanding regarding interaction between pathogens and human exposure through to epidemic spread are also highlighted. Finally, probable future prospects of research in the field of medical science concerning the global pandemic of SARS-CoV-2 are also highlighted.

## 2. Overview of SARS-CoV-2 and Its Pathogenic Effect on Humans

SARS-CoV-2 is at the center of worldwide panic and is a global health concern since December 2019. According to recent WHO reports, about 173,005,553 total cases with 3,727,605 fatalities (updated 8 June 2021) [23] were reported globally and the number is increasing in an intimidating way every day [24,25]. WHO designated the disease a very high-risk category due to the rapid leap in the total number of confirmed COVID-19 infected persons and affected countries. According to WHO, a total 222 countries are affected by the intimidating SARS-CoV-2 pandemic to date. At the time of writing, the number of active cases was found to be high in many countries, India being the second most affected country due to its higher population. Because India experiences different climate conditions, it faced a first and second wave of COVID-19 as the virus multiplied. This resulted in the total active cases in India being more than 28,900,000 deaths (3.49 L), with a mortality rate of 44%. In addition, the number of patients recovering (27,159,180) also decreased because the effectiveness of the new corona variant is much higher than before. At the time of writing, there is a decrease in the mortality rates resulting in fewer active cases, whereas updated COVID-19 active cases and deaths in India, according to WHO, were reported to be 43,100,000 and 524,000, respectively.

The India COVID-19 update with statistics and graphs is shown in Figure 2 (dated 8 June 2021). Based on the data from the initial cases in Wuhan and investigations conducted by the China Center for Disease Control (CDC) and local CDCs, the usual incubation time (i.e., the period after exposure and before having symptoms) was reported to be within 3 to 7 days (median 5.1 days, similar to SARS and up to 2 weeks as the longest time from infection to symptoms being 12.5 days [26,27]. These data also proved that this novel pandemic doubled about every seven days, with the basic reproduction number being 2.2. In other words, on average, each patient transmits the infection to an additional 2.2 individuals [28]. The initiation of the infection process by the SARS-CoV-2 virus is the invasion of lung type II alveolar cells via a receptor protein called angiotensin-converting enzyme 2 (ACE2) present on the cell membrane with a glycosylated spike (S) viral protein that medi-

ates host cell invasion, as schematically represented is discuss by Zargar et al., 2020 [29]. The COVID-19 virus spreads primarily through small droplets of saliva or discharge from the nose or mouth, when a COVID-infected person sneezes, coughs or speaks. These discharges are relatively heavy so they do not travel far, and quickly sink to the ground or other surface. People can catch the virus by breathing in these discharged droplets from the virus-infected person. Older people or those with existing chronic medical conditions, such as heart disease, lung disease, cardiovascular disease, diabetes, severe obesity, chronic kidney or liver disease, chronic respiratory disease, cancer or low immune systems may be at higher risk of developing severe illness [23,30].

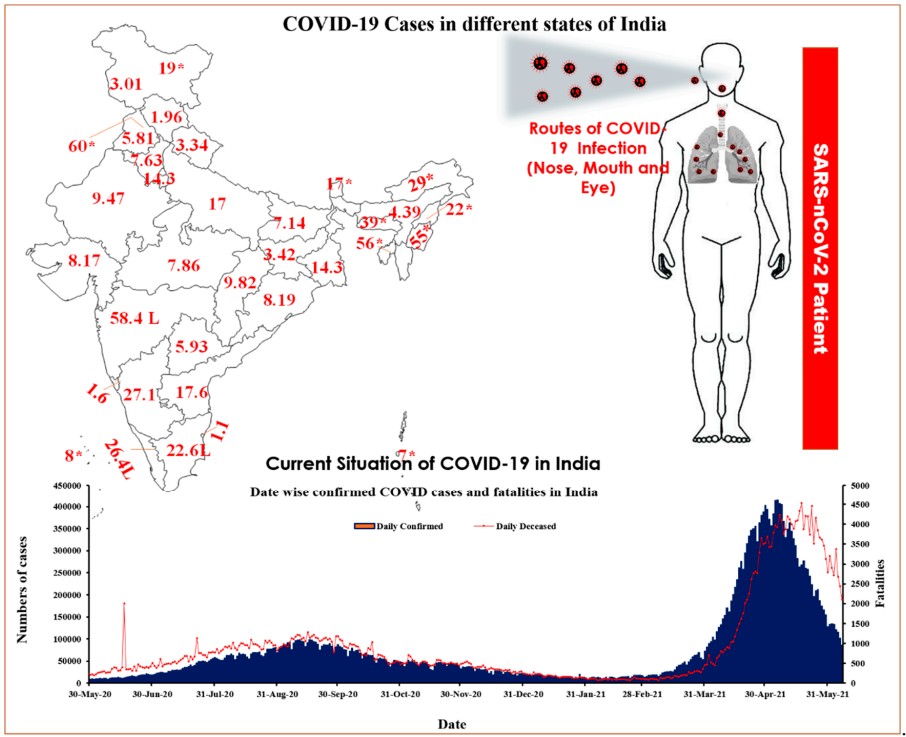

**Figure 2.** Statistics and graphical presentation of COVID-19 pandemic across India (8-June-2021).

This is similar to that seen with other respiratory diseases, such as influenza. Weiss and co-workers [1] recently confirmed the current taxonomic classification of the 2019-nCoVs through alignment-free comparative genomics with the help of the information correlation and partial information correlation (IC–PIC) method. They reported that the evidence supports classification of 2019-nCoV as a sarbecovirus sub-genus within beta coronavirus. The acquired IC–PIC tree for beta coronavirus supports the supposition that the mentioned novel viruses are of bat origin with pangolin as one of the possible intermediate hosts. Furthermore, the IC–PIC tree of 2019-nCoVs provides details of the disease spreading and also provides evidence for searching for the origin of the outbreak in Wuhan [31]. A flow-diagram showing the pathways for SARS-CoV-2 is shown in Figure 3. According to health experts, this novel coronavirus originated from bats or perhaps via a different animal species in late 2019 in a food market in the Chinese city of Wuhan, where wildlife was illegally sold, and then passed to humans [32]. Recently, it was seen that SARS created international apprehension due to its novelty and communicability, as well as its rapid spread through jet travel and a massive proportion of the exposed medical and nursing personnel being infected. A tragic fact about COVID-19 is that some people may have COVID-19 and spread it to others, even if they do not have symptoms or do not know they have COVID-19. Studies reveal that the survival duration of the COVID-19 virus is up to 72 h on plastic and stainless steel, less than 4 h on copper and less than 24 h on cardboard in any climate conditions.

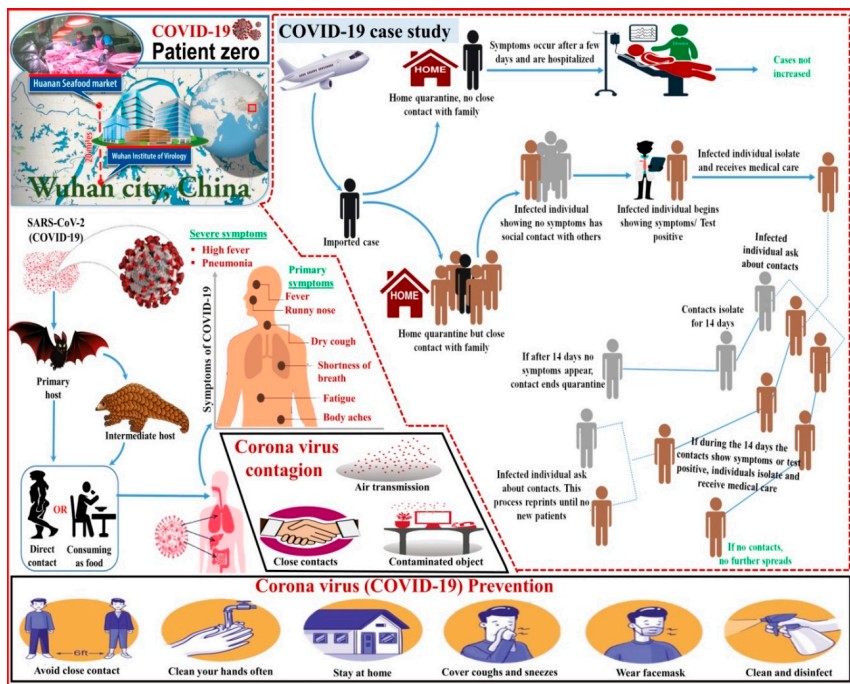

**Figure 3.** Flow-diagram showing the pathways for human coronavirus (SARS-CoV-2) and case study including person to person transmission methods along with the recommended precautions.

Cleaning hands with an alcohol-based hand rub or washing them with soap and water for at least 20 s are some primary preventive measures against the virus [1,9,32,33]. Washing hands with soap and water is an effective way of destroying and dislodging the novel SARS-CoV-2, as shown in Supplementary Tables S1 and S2. Even though there are few vaccines and drugs available to prevent COVID-19 presently, one can always take steps to reduce risk of infection. Therefore, to avoid COVID-19, WHO and CDC recommend several precautions, displayed in Figure 3 above. Driggin et al. reported that this infection may directly be the cause of, or affect, certain cardiovascular diseases (CVD) [34]. In other words, pre-existing CVD may prejudice a person's immunity to be prone to COVID-19 infection. Therefore, it was observed that those already suffering with CVD and found to be infected by the virus have an elevated risk of adverse outcomes as the infection itself is allied with CV complications. Furthermore, the COVID-19 infection may also have direct as well as indirect effects pertinent to a person's CV health. In a chronology that includes the current date, an epidemic of cases with inexplicable low respiratory infections detected in Wuhan was first reported on 31 December 2019, to the WHO Country Office in China. In their recent article, Schroder and co-workers [35] compared the onset of the COVID-19 outbreak toa looming storm and also discussed the world's inability to envisage the extent of the storm's severity despite the availability of data.

The available literature traces the beginning of symptomatic individuals to the commencement of the month of December in 2019. As the scientists could not identify the causative agent, these first reported cases were categorized as "*pneumonia of unknown etiology*". The CDC and local CDCs organized a rigorous outbreak investigation program and accredited the etiology of this illness to a novel virus belonging to the coronavirus (CoV) family [36]. Ultimately, WHO, announced on 11 February 2020, that the disease caused by this new CoV was "COVID-19," which is the contraction of "*Coronavirus disease 2019*". COVID-19 is the third CoV outbreak in humans that is said to have transpired in the last two decades, causing clinical symptoms of respiratory, digestive and systematic affections, majorly articulated by pneumonia [37,38]. It is noteworthy that in the last two decades, two supplementary CoV epidemics have been reported, namely, SARS-CoV, which aggravated a large-scale epidemic commencement in China and the MERS-CoV, which began in Saudi Arabia and is still said to cause infrequent cases. Sadly, the new virus appears to be highly

contagious and has quickly spread worldwide. At the time of writing, the therapeutic strategies to deal with the infection are only supportive and preventative, aimed at reducing transmission in the community and considered as helpful tools since no successful vaccine has been developed so far. It was seen that aggressive isolation measures in some countries led to a progressive reduction in cases. As mentioned above, transmission of this disease is said to occur by means of respiratory droplets (>5–10 μm diameter) released during the coughing and sneezing from infected humans. The knowledgeable carriers can shed the virus fora period of up to 21 days to other people coming in contact to them [37]. The meticulous depiction of the desquamation of pneumocytes, formation of hyaline membrane, bilateral diffused alveolar damage and presence of cellular fibromyxoidexudate was detailed after histological examination, performed on biopsy tissues obtained from lung, liver and heart tissue. As marked cytopathic effects, multinucleated syncytial cells, atypical enlarged pneumocytes and interstitial mononuclear inflammatory infiltrates with the majority of lymphocytes in the affected lungs are prominent features [39]. It is noteworthy that aerosol transmission is also a possibility in the case of prolonged exposure to prominent aerosol concentrations in closed spaces as reported [40–42]. Analysis of data related to the spread of SARS-CoV-2 in China pointed out that close contact between individuals is necessary for transmission. According to Zhang et al., the transmission or spread of the virus is mainly limited to family members, healthcare professionals and other close contacts (in a diameter of about 6 feet or 1.8 m) persons [43]. A worth mentioning, yet intimidating, fact is that the pre- and asymptomatic individuals may contribute to about 80% of COVID-19 transmission. It was mentioned by Bai et al., 2020 that if the presumed transmissions by an asymptomatic carrier were found to be replicating, then the prevention of COVID-19 infection would be challenging [44], which proved to be correct, indeed. Furthermore, it seems that the chances of contamination are higher in intensive care units (ICUs) than in general wards, with SARS-CoV-2 being found on floors, computer mice, trash cans, and sickbed handrails, as well as in the air up to a diameter of 4 m from the patients [36]. However, it is noteworthy that infection with the pathogen does not compulsorily lead to disease. Infection occurs when viruses, bacteria or other microbes enter the human body through the mouth, nose and eyes. Many drugs are specifically designed to kill microbes or inhibit further growth within the host environment [21,45]

## 3. Challenges and Burden of the COVID-19 Pandemic

In this section, we discuss socio-economic, health and national and international healthcare challenges following lockdown, with a focus on the population belonging to the low socio-economic stratum (SES). The social fabric of India thrives on interdependence, both economic and emotional, within families, relatives and friends [46]. Close contact, such as living in crowded housing and other places, pushing and jostling are extremely ordinary and are a deterrent to social distancing as dictated during this present pandemic. Despite the lockdown, crowding has been observed in religious places, during travel or by migrants on buses, trains, airplanes, local and private vehicles, and even while purchasing liquor at the shops [47]. The more troubling aspect is the lack of proper provision of food safety for those hit the hardest by lockdown: due to the enormous scale of the problem, the government schemes remain vastly inadequate [47]. As a result of the lockdown, there is an increased possibility of malnutrition and unemployment among those in the low SES. On the other hand, due to the lockdown situation, such people are becoming more affected, as they earn and feed their families on a daily basis. According to Indian government guidelines, the Food Corporation of India recently allotted 1,296,000 metric tons of food grains under the Pradhan Mantri Garib Kalyan Anna Yojna (PMGKAY) as Government of India initiative in its fight against COVID-19 [48]. The efficacy of this scheme and the adequacy of food distribution remain to be seen.

The COVID-19 lockdown situation is particularly stressful to human beings because it is hard to predict how the situation will develop, and their circumstances can rapidly alter [49]. This can leave people feeling powerless, as though they are no longer in control

of their own lives. In many areas of their lives and in this situation, there are many aspects individuals cannot control, such as actions and reactions of many other people, how long the situation will last, and what might happen in the future. Problem-solving skills can also help people manage their thoughts and worries during this situation. Such skills allow person to define exactly what they are worried about, and then find the best solution to prevent catching COVID-19. At the family level, the pandemic has led to a re-organization of everyday life to cope with the stress of quarantine and social distancing. Parents have experienced increased pressure to work from home and to keep jobs and businesses running, as well as to taking care of children at home and home-schooling them at the same time, while caregiver resources, including grandparents and the wider family, have been restricted [50]. Grief and mourning for lost family members, especially in cases where contact with the infected member was restricted or refused, could lead to adjustment problems, post-traumatic stress disorder, depression and even suicide of both adults and younger people [50]. All these facts indicate that the prevention of this currant corona pandemic is a major challenging task for everyone, including ordinary people, scientific workers, researchers, medical teams and other sociological and ecological communities. Some of the points are:

- There is an increased possibility of malnutrition and unemployment during lockdown.
- A weakened physical condition increases a person's susceptibility to disease.
- With the weakening of the socio-economic system, a large number of people may lose their livelihood and will be in danger of irreversible impoverishment.
- Public messages encouraging frequent handwashing can put such people at risk and exacerbate their mental illness.
- The use and subsequent irresponsible disposal of face masks in large numbers leads to a risk of soil and water pollution, which can result in harmful effects on humans and animals.

Screening for the potential for possible SARS-CoV-2 infection is based on knowledge of the characteristics of clinical illness observed in the early cases, and the geographic distribution of current cases [51] Screening reflects the current public health goal of rapidly containing and preventing transmission of SARS-CoV-2 globally. The assessment is intended to allow a healthcare provider to make decisions about appropriate infection control and the management of patients. Note that the signs and symptoms of SARS-CoV-2 are similar to those associated with other viral respiratory tract infections. A flow diagram of potential COVID-19 case management from infection to discharge is shown in Figure 4. It is a big challenge for researchers and virologists to find a solution for the deadly COVID-19. According to WHO, this is attributed to the fact that COVID-19 is a viral infection known to have the fastest frequency of recombination or replication in its positive strand, resulting in the quick formation of new progeny viral cells inside the host cells, as compared with other viruses [52]. The genomic structure of the virus is not the only factor that presents a great challenge to research; its ability to adapt and survive in different environmental conditions make it nearly impossible to identify its mode of survival [22,52]. It was earlier reported by Tanne et al., 2020 [53] that the SARS virus can survive at 4 °C with a humidity rate of 20%. The first outbreak of the SARS-CoV-2 was during the peak of winter, when the environmental temperature was around 2 °C to 10 °C, but since then the virus has infected people and survived in countries with completely different climatic conditions, making its demographic association impossible to predict [28,52,53]. The harmful viruses, namely CoVs, SARS and MERS, which caused fatal outbreaks earlier, constantly arise; at the time of writing, the budding COVID-19—initially expressed as pneumonia of unknown origin—thus presents a remarkable threat to community health across the world [54–56].

Because of the earlier threats of SARS, MERS, and of the current COVID-19 situation, it is necessary to design campaigns and argue against emerging and zoonotic pathogens that could pose pandemic threats and place human lives at risk [57,58]. (Even though sequences of some clinical trials are now ongoing (at the time of writing) and others completed to identify the potential therapies or vaccines to respond to SARS-CoV-2, no completely successful attempts were made to date.

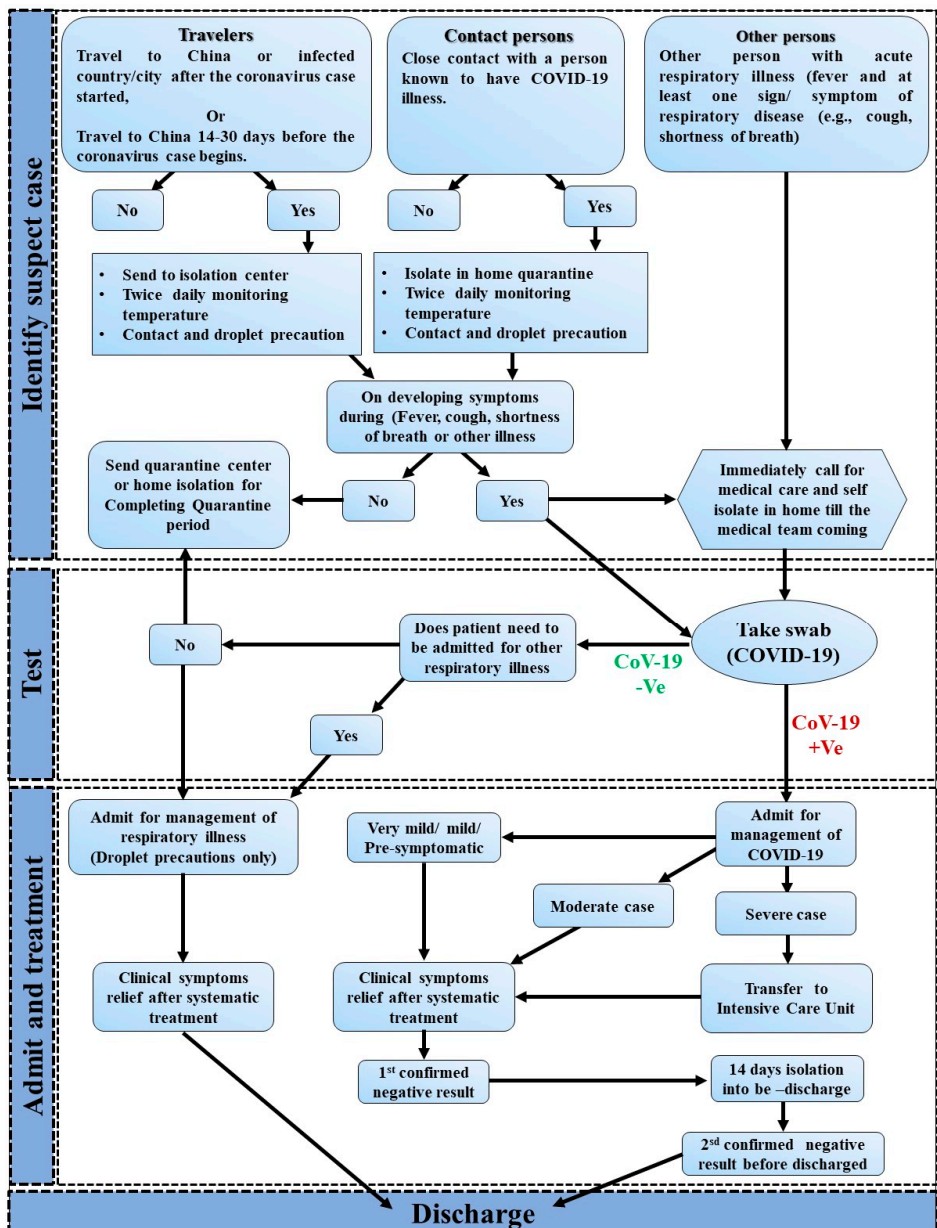

**Figure 4.** Flow diagram of potential COVID-19 case management with respect to identifying suspected cases, test, admit, treatment and discharge.

## 4. Treatment Strategies and Clinical Development of COVID-19 Vaccine: A Global Solution and Future Perspectives

Recommended different classes of drugs for treatment of COVID-19 are listed in Table 1. Few of the reports mention the possibility of probable pandemic risks and threats. The potential for rapid increases in cases of COVID-19 is galvanizing us to make strenuous efforts to check the spread of this widely circulating virus among world population by following appropriate prevention and control measures, along with formulating global solutions and modified strategies with future perspectives [59–63]. At present, scientists and physicians around the world are competing to understand this human CoV, along with its patho-physiology, in order to uncover potential treatment regimens, and also discover certain effective therapeutic drugs and vaccines [62]. Zhang and co-workers reported that molecular tools are widely preferred for timely diagnosis of COVID-19 [43]. At the peak of the epidemic, serological diagnosis was not a good deal of help, while recovered-patient serum samples can be tested to know the titer of IgG.

**Table 1.** Different class of drugs for treatment of COVID-19, sources, treatment strategies, routes for working and efficiency.

| Drug Name | Class | Source | Risk Factor | Routes for Working | Overall Efficacy | Ref. |
|---|---|---|---|---|---|---|
| Hydroxy-chloroquine | Antimalarials | Bioactive compound | Retinal injure, cardiac arrhythmias; G6PD deficiency patients; Caution in patients with diabetess; Noteworthy drug interactions | Inhibition of viral enzyme or processes of RNA polymerase, viral DNA, virus assembly, new virus particle transport and virus release; In vitro activity against SARS-CoV-2; ACE2 cellular receptor inhibition, acidification at the cell membrane inhibiting of the virus | In vitro and restricted clinical data suggest potential advantage | [64,65] |
| Chloroquine | Antimalarials and amebicides | Bioactive ompound; or phytochemical Extracted from the plant Artemisia annua (also known as"Sweet annine") | Retinal injure, cardiac arrhythmias; G6PD deficiency patients; Caution in patients with diabetes; Noteworthy drug interactions | Inhibition of viral enzyme or processes of RNA polymerase, viral DNA, virus assembly, new virus particle transport and virus release; In vitro activity against SARS-CoV-2; ACE2 cellular receptor inhibition, acidification at the cell membrane inhibiting of the virus | In vitro and restricted clinical data suggest potential advantage | [66] |
| Remdesivir | Antiviral | Chemically synthesized | Efficacy will require ongoing randomized, placebo-controlled trials of remdesivir therapy; Remdesivir has broad-spectrum activity against members of several virus families, including filoviruses; | Acts as an inhibitor of RNA-dependent RNA polymerases; Acts as a broad-spectrum antiviral with in vitro activity against CoVs; Once incorporated into the viral RNA at position i, RDV-TP terminates RNA synthesis at position | Investigational and available only through expanded contact and study protocols; numerous clinical trials are underway | [64,65] |
| Tocilizumab | Interleukin-6 (IL-6) Receptor-Inhibiting Monoclonal Antibody | Humanized (from mouse) | Hepatatoxicity; Neutropenia and thrombocytopenia; Risk of GI perforation and infusion-associated reactions | Inhibits IL-6-mediated signaling by competitively binding to both soluble and membrane-bound IL-6 receptors. IL-6 is a proinflammatory cytokine that is involved in diverse physiological processes; IL-6 is produced by various cell types, including T and B cells, monocytes, lymphocytes and fibroblasts; Immunoglobulin secretion induction, hematopoietic precursor cell proliferation, hepatic acute-phase protein synthesis initiation; Cytokine release condition may be a component of severe sickness in COVID-19 patients | Limited preliminary data as adjunct therapy; Immunomodulating negotiator used in some protocols based on theoretical mechanisms | [67] |
| Azithromycin | Macrolide Antibiotics (Antibacterial) | Chemically synthesized (Semi-synthetic) | Risk of cardiac arrhythmias; Noteworthy drug connections | It may have immunomodulatory properties in pulmonary inflammatory disorders; They may downregulate inflammatory responses and decrease the excessive cytokine production related torespiratory viral infections; It may prevent bacterial super infection, and macrolides may have immunomodulatory properties to work as adjunct therapy | Used in some protocols based on theoretical mechanisms and limited preliminary data as adjunct therapy | [64] |
| Honey, Food substance | Hymenopterans | Bioactive compound | Still needs to be proved through clinical trial and proper experiments | The COVID-19 infected individual having cytokine syndrome can be tackled with honey antioxidant property and increased IFN-γ level. | Polyphenol-rich environment can efficiently activate local immune suppression and tissue repair mechanisms; Investigational use is being studied; | [68] |
| COVID-19 convalescent plasma | Antibodies | Plasma collected from persons who have recovered from COVID-19 that may contain antibodies to nCoV-19 | Awaiting as clinical trials are ongoing; Corticosteroid administration issues; Cardiac arrest problem | Clinical trials are being conducted to evaluate the use of COVID-19 convalescent plasma to treat patients with severe or immediately life-threatening COVID-19 infections. COVID-19 convalescent plasma is not intended for prevention of the infection; Corticosteroid therapy is not recommended for viral pneumonia; Acute respiratory distress syndrome | Not recommended for viral pneumonia; but, use may be measured for patients with acute respiratory distress condition | [65] |

According to the diagnosis and treatment program (*6th, 165 version*) published by the National Health Commission (NHC) of the People's Republic of China, the diagnosis of viral pneumonia was defined on the basis of radiologic features used as diagnostic criteria for SARS-CoV-2 (NHC, 2020). Numerous attempts were made to develop vaccines against human CoV infection in past decades, some vaccines providing a degree of cross-protection, but these are greatly limited owing to the extensive diversity in antigenic variants within the strains of a phylogenetic sub-cluster [69]. As for SARS and MERS-CoVs, there are few approved definite antiviral treatments or vaccines accessible till date. Table S1 displays the developed COVID-19 vaccines, and their origins, by different regulatory agencies. On the other hand, few of the advances made in developing therapeutics and vaccines for SARS and MERS-CoV could be adopted for countering COVID-19. Because efforts to propose and develop any vaccine or antiviral mediator to tackle the presently up-and-coming CoV pathogen would take some time, it is necessary to rely extensively on enforcing extremely effective preclusion and control measures to diminish the COVID-19 transmission and spreading danger [70]. As contaminated individuals are hospitalized, individual patients should immediately be given symptomatic and sympathetic treatment as per the severity of their symptoms [71,72]. Therapy may comprise high-flow nasal cannula (HFNC) oxygen or supplementary oxygen therapy through the nose to reduce breathing anxiety; if needed, methylprednisolone could be administered intravenously to correct hypoxemia. In extreme cases, adrenaline and repurposed drugs, such as lopinavir plus ritonavir as an anti-viral drug, and moxifloxacin or another antibiotic, may be needed; these could be administered intravenously (IV) and orally, respectively, to prevent secondary bacterial infection [73]. Lopinavir andritonavir belong to the anti-retroviral class of drugs with a protease-inhibitor action; these are widely employed in the treatment of the human immune virus (HIV), as reported by [71]. In recent times, combined lopinavir/ritonavir was suggested as a promising candidate in treating SARS-CoV-2 infection [74,75]. Nevertheless, multiple drugs are under study, including other antiretrovirals such as remdesevir, and also antivirals such as oseltamivir and other therapies, including chloroquine and even indomethacin. Remdesivir is a novel nucleotide analog prodrug that was intended to be used for the treatment of Ebola virus disease and this study was reported by Sheahan et al., 2017 [76]. It also has anti-coronavirus activity due to its inhibitory action on SARS-CoV and MERS-CoV replication [77,78]. Other drugs, such as baricitinib (authorized by the WHO), the human recombinant monoclonal antibodies casivirimab/imdevibad and the antiviral molnupiravir (by the VHO and the FDA) were also included for treating SARS-CoV-2 infection.

At present, efforts are being made to identify and develop monoclonal antibodies that are specific and effective against SARS-CoV-2. The majority of the vaccines being developed for CoVs target the spike glycoprotein or S protein [69]. This is mainly because S protein is the most important inducer of neutralizing antibodies [79]. Several kinds of antiviral drugs and vaccines based on the S protein were previously evaluated. Among them, the S protein-based vaccines include full-length S protein vaccines, viral vector-based vaccine, DNA-based vaccine, recombinant S protein based and recombinant RBD protein-based vaccines. On the other hand, S protein-based antiviral therapies include RBD–ACE2 blockers, S cleavage inhibitors, fusion core blockers, neutralizing antibodies, protease inhibitors, S protein inhibitors and small interfering RNAs [80]. Even though such therapeutic options have proven efficacy in in-vitro studies, most of them did not undergo randomized animal or human trials and, hence, are of limited use in our present COVID-19 scenario. While Veterinary quarterly 73 might be considered as the ideal therapeutic option for COVID-19 [60], further evaluation is required before confirming the efficacy of such combination therapy. A variety of different therapeutic and vaccine design approaches against CoVs are being explored and evaluated due to their good potency, efficacy and safety; hopefully the process of evaluation will be accelerated in the near future because researchers are making strenuous efforts to design and develop suitable vaccines for COVID-19 [66,81–84]. In this situation, professional monitoring and treatment of COVID-19 pneumonia using active prevention and control measures and following

the international and national urbanized guidelines in a scientific manner are of utmost importance [31]. Vaccines based on recombinant SARS-CoV-2 S protein subunit-trimers are also in the development pipeline [29]. DNA-based vaccine INO-4800, developed by Inovio Pharmaceuticals, has entered Phase I trials in the US [31]. The six other vaccines, based on the SARS-CoV-2 antigens in the phase I trials, include adenoviral vector 5 (NCT04313127), lentiviral vector (NCT04276896), artificial antigen-presenting cells or a APC (NCT04299724), mRNA (NCT04283461) and chimpanzee adenoviral vector ChAdOx1 (NCT04324606); the latter, developed by University of Oxford, showed promising results in monkeys to fend off the deadly virus with no adverse effects [85]. Previously, a vaccine was required to pass three separate clinical trials in order to progress from laboratory to clinic. Prior to the first clinical test, animal testing is performed, for example on pigs and mice, to assess immune response and this process is called a "pre-clinical test". After passing the pre-clinical test, the vaccine is used on a small group of people to verify whether it has any adverse effect on the immune system. This is referred to as Phase I of the clinical trial. In the second phase (Phase II) of a clinical trial, the vaccine must pass another safety test on a relatively large set of people. It is tried on hundreds of people divided into small groups, while in the third phase (Phase III), also known as the "efficacy test", the vaccine is used on thousands of people. After the third phase, health regulators of respective nations review the results and approve it for public use accordingly [86]. During the COVID pandemic, however, the procedures were extraordinarily shortened, and mixed phase II and III studies were carried out in parallel; permission was also obtained for application to patients with their informed consent on the limited information available.

Recently, the latent benefit of chloroquine/hydroxychloroquine (CQ/HCQ) in the treatment of the newly emerging virus pandemic gained major attention [64,65]. CQ is basically an amine acidotropic form of quinine, known for decades as a vanguard drug prescribed for the treatment and prophylaxis of malaria worldwide. HCQ is a 4-aminoquiniline analogue of chloroquine, which has a hydroxyl group at the end of aside chain of CQ; its sulphate form is currently obtainable for oral administration with a higher dose than CQ for quick gastrointestinal absorption, as well as renal elimination [64]. It is noteworthy that the pharmacokinetic features of HCQ were found comparable with CQ so it can be used for long periods with an improved tolerability [64,65]. Disappointingly, it was seen that the usefulness of CQ/HCQ gradually decreased due to the increasing emergence of the chloroquine-resistant plasmodium falciparum strains. The range of its antiviral effects and clinical response to SARS-CoV merit a particular attention for repurposing its use in the treatment of the nCoV infection. The Indian Council of Medical Research (ICMR), under the Ministry of Health and Family Welfare, recommended the chemoprophylaxis and therapeutic applicability of HCQ in SARS-CoV-2 treatment with a dosage of 400 mg twice on day 1, then 400 mg once a week thereafter for asymptomatic healthcare workers treating infected patients, as well as for the asymptomatic household contacts of confirmed cases [87].

Indian medicinal herbs are a promising field for the treatment of various illnesses [88]. Ayurveda and Siddha practices originated in India and are still widely used among the Indian population. Many researchers and scientists identified certain phytocompounds to effectively characterize medicinal herbs that could help to alleviate the infection [89,90]. Hence, by repurposing Indian medicinal plants, more innovative treatment options can be logged for their role in defeating this viral transmission. At a time of worldwide anxiety, it is imperative to find long term solutions to prevent the transmission of such pandemics. It is, therefore, time for all the citizens to join hands together to fight against the coronavirus by practicing self-hygiene and social distancing [86,89]. Pal et al. (2020) describe a broad spectrum of antioxidant activity in honey; it was suggested that honey could possibly act as a protective agent for infected patients with viruses, such as influenza or coronaviruses; however, this still needs to be proved through clinical trials and proper experiments [90]. ACOVID-19-infected individual with cytokine syndrome can be treated with honey's antioxidant properties and with an increase in IFN-γ level. Various researchers have found that honey may act as a preventive agent against hyper inflammation caused by

SARS-CoV-2 [91,92]. In a case study, it was observed that a polyphenol-rich environment can efficiently activate local immune suppression and tissue repair mechanisms. Honey is also rich in bioactive compounds, hence, it can be concluded that it may have a possible role in alleviating pain in COVID-19-infected patients, but this needs further clarification through clinical trials and properly designed experiments, as stated by Correa et al. (2019) [68]. It might, therefore, be hypothesized that honey might be useful for COVID-19 patients because of several major mechanisms, such as direct virucidal properties, regulating or boosting host immune signaling pathways and improving comorbid conditions. In addition, based on the previous results of several studies, honey may act as a preventive agent against hyper inflammation caused by SARS-CoV-2 [93].

## 5. Methods Mainly Employed for COVID-19 Detection in Human

Presently, several main methods are employed for COVID-19 detection or monitoring in humans:

- *Rapid antigen test (RAT) method*,
- *Reverse transcription polymerase chain reaction (RT-PCR) method*,
- *Computed tomography (CT) imaging method*,
- *CXRs imaging method*.

The rapid antigen test (RAT), or COVID-19 lateral flow test, is employed in the detection of SARS-COV-2 infection and used for the analysis of respiratory pathogens. Q-COVID-19 Ag detection kits are used for inherent COVID antigen test, as well as a sterile swab for sample collection (WHO, 2020) from infected individuals. In this test, a human nasopharyngeal swab is used to quantitatively detect specific protein fragments, called antigens, which exist in the virus. Two marked lines, namely the control line and the test line, are present on the top of a nitrocellulose membrane in the test kit (online, 2020). At first, these lines are not visible; however, if the SARS-CoV-2 antibody is present in the applied specimen, a colored line appears. The intensity of the color of this line expresses the amount of viral antibody present in the test sample. This test was established as a helpful tool in identifying further outbreaks of the virus, and also assisted in significantly reducing the mortality rate. This test requires no area specifications and can be performed anywhere, for instance in clinics, offices and even in remote locations. In addition, the results can be obtained in around 15 min and administration of the test requires minimal training, thereby offering cost-effectiveness compared with a RAT. In brief, the method proves to be cheaper with easy handling and rapid results.

Reverse transcription polymerase chain reaction (RT-PCR) is a major laboratory technique that combines reverse transcription (RT) of RNA into DNA and amplification of specific DNA targets employing polymerase chain reaction (PCR), hence the name RT-PCR [94]. For the detection of SARS-CoV-2 using RT-PCR test, the target includes gene encoding (N, E and S proteins), RNA-dependent RdRP gene and open reading frame lab [94]. This is chiefly employed in the measuring of the specific amount of RNA due to the conserved beta CoVs, the E genes and N genes cross reacting with CoVs [84]. SARS-CoV-2 can be differentiated from SARS-CoVs by employing the RdRP gene. In addition, it can also be differentiated using the S gene because the S gene is extremely divergent from other CoVs [95]. Seven RT-PCR assays were developed by scientists from all over the world and were quickly made available for COVID-19 diagnosis by WHO. Wu et al., 2020 and Zhou et al., 2020 [95,96] performed early identification and specific sequencing of the SARS-CoV-2 genome that facilitated the rapid development of the RT-PCR techniques. At the beginning of pandemic, using RT-PCR assays, presumptive cases were recognized and confirmed with genomic sequencing. In comparison with RT-PCR, genome sequencing is pricier and time consuming. Therefore, current molecular diagnosis for COVID-19 is first and foremost based on the RT-PCR viral RNA detection technique. The first real time RT-PCR assays from Germany, targeting the RdRP, E and N genes of SARS-CoV-2 was published on 23 January 2020 and produced the greatest analytical sensitivity, i.e., 95% detection probability. The use of the N gene as the target of the same resulted in ten times

more sensitivity than the open reading frame lab for SARS-CoV-2 detection [94]. According to their needs, scientists can choose the different targets for RT-PCR assays.

Challenges for improvements in molecular diagnosis also exist in developing new droplet digital RT-PCR platforms, such as better compatibility with simpler sample handling methods and improved tolerance of matrix effects, etc. as compared with other detection methods [97]. RT-PCR is used for the same primers, probes and reagents as for the conventional RT-PCR technique, except that the bulk reaction solution is divided into thousands of nanoliter-sized microdroplets [97]. There are two attractive features of droplet digital RT-PCR, providing motivation for further development. Firstly, both these features may result in lower detection limits i.e., the partition efficiently reduces template competition for primers. Secondly, the nanoliter volume of the isolated droplet reactors dramatically enhances the local effective concentration of the target, recommending reaction kinetics and efficiency [41]. The process of RT-PCR assays normally completes within a duration of about 1 to 3 h. Therefore, this technique is very fast and sensitive, and uses a very small size of sample (nL) compared with others for the detection of SARS-CoV-2 virus in humans.

In COVID-19 diagnosis, computed tomography (CT) or more specifically computerized X-ray imaging can be employed as an essential complement to RT-PCR in the current epidemic context [62,98]. There is a long interval between the commencement of symptoms and the disease being observed using CT images. In the initial state (first 2 days), COVID-19-positive patient lungs usually show as normal on the CT image. In addition, it was observed that from the lung CT study of patients, the maximum of lung disease can be observed after 10 days of symptoms beginning [99]. A wide variety of CT was reported in different studies on the detection of COVID-19 in 2020. In countries such as Turkey, due to the lack of availability of test kits at the onset of the pandemic, CT imaging was widely used for COVID-19 detection. Similarly, in Chinese clinical centers, there was an insufficient number of test kits and these had a high rate of producing false-negative results at the onset of COVID-19 symptoms, so doctors at Chinese clinics encouraged diagnosis based on clinical and chest CT [96]. Because of the importance chest CT, it is important for radiologists to become familiar with classic CT associated with the new task, and also the imaging criteria for an alternative diagnosis. The new CT devices emit little ionizing radiation, little more than that of two chest X-rays, so it is therefore a very safe procedure. However, CT scans make use of X-rays and it is a known fact that X-rays produce radiations which are ionizing. These ionizing radiations may result in adverse biological effects in the tissues of living beings and the risk of these effects accumulates with every exposure.

In terms of the initial imaging test of patients suspected with COVID-19, CXRs remains the first choice as the radiation dose of CXR is lower than that of chest CT scans, i.e., 0.02 mSv and 7 nSm, respectively, which can reduce the risk of radiation-related diseases, such as cancer, to patients [100,101]. In addition, from the financial point of view of healthcare systems and patients, CXR is cheaper than CT scans. Portable CXR units can be wheeled into an emergency room and ICU wards can be easily cleaned. Other attempts at COVID-19 prediction by means of medical imaging were undertaken; however, CXR results were reported for a private data set and are not reproducible [101] and many researchers were represented without appropriate assessment, potentially leading to over-estimation and over-fitting [52]. For the detection of COVID-19, the commonly reported CXR and CT include ground-glass opacities and lung consolidation. The CT findings of ground-glass densities may correlate with the difficulty of finding the disease on CXR [102].

## 6. Comparative Study for SARS-CoV-2 and Others Virus to Effect on Human

It is a fact worth mentioning that the human sera collected before the SARS outbreak did not contain antibodies directed against SARS-CoV, signifying that this virus is new to humans [25,45]. The complete 29,727-nucleotide sequence of the RNA genome of SARS-CoV-2 (Gen Bank accession numbers AY274119 and AY278741) confirm that the novel CoV is a member of the *Coronaviridae* family and offers some insight into its probable

origin [103,104]. The genes of SARS-CoV-2 were evaluated against the corresponding genes of the known coronaviruses of humans, pigs, cattle, dogs, cats, mice, rats, chickens and turkeys, and it was concluded that each gene of SARS-CoV-2 has only about 70% or less identity with the corresponding gene of the known coronaviruses. Consequently, the SARS-CoV-2 is only faintly related to the known coronaviruses of humans and animals. SARS-CoV-2 is enzootic in an unknown animal or bird species and was genetically isolated there for a relatively long time before somehow suddenly rising as a noxious virus of humans [22]. It is noteworthy that the SARS-CoV-2 and Ebola, Hendra and Nipah viruses all belong to variant viral families and are identified causative agents of lethal viral diseases. Elshabrawy and co-workers [105] reported that these viruses are dependent on cathepsin L for their entry into the target cells. It is seen that infections in humans with the human CoV strains HCoV-229E, HCoV-OC43, HCoV-NL63 and HCoV-HKU1 typically result in mild, self-limiting upper respiratory tract infections, such as the common cold. While on the other hand, the CoVs responsible for SARS and MERS received global attention because of their capability to cause community and healthcare-associated outbreaks of severe infections in human populations. The zoonotic viruses, including the Ebola virus (EBOV), Hendra virus (HeV) and Nipah virus (NiV) are classified as BSL-4 pathogens due to their previously described ability to cause severe illness, or even death, in humans due to no authentic vaccines having been developed against them. SARS-CoV-2, Ebola virus (EBOV), Hendra virus (HeV) and Nipah virus (NiV) are highly infectious in nature [106,107]. They were specified with various species of bat as their identified natural reservoir and are also the causative agents of SARS along with the severe acute hemorrhagic fever (EBOV), as well as the lethal encephalitides (HeV and NiV). The enveloped viruses enter their target cells by means of fusion of the viral envelope with the host cell membrane, thereby delivering their viral genome to the cytoplasm of the target sell. The most intimidating fact about SARS-CoV-2 in comparison with all the other zoonotic viruses is its mortality rate [107,108]. A total nine host miRNAs with a potential to target SARS-CoV-2 genes were identified; namely, hsa-let-7a, hsa-miR101, hsa-miR125a-5p, hsamiR126, hsa-miR222, hsa-miR23b, hsa-miR378, hsamiR380-5 and hsa-miR98. However, these nine identified miRNAs do not have targets in genomes of SARS and MERS, which are close relatives of SARS-CoV-2, but were reported to target the genes of certain other viruses, such as Herpes simplex virus 1, Hepatitis C, Hepatitis B, Influenza A and vesicular stomatitis virus [109]. Table 2 represents the parallel investigation of SARS, MERS, COVID-19 and Common Flu with their symptoms, onset of disease, incubation period, recovery and transmission of disease, along with complications and available treatments. The symptoms of COVID-19 disease caused by the new CoV overlap with those associated with common colds, cough, allergies and influenza that can make it difficult to diagnose. The CoV primarily affects a patient's lungs and commonly causes fever, dry cough and shortness of breath. In this review, all symptoms associated with COVID-19 and comparisons with the common cold, cough, allergies and influenza are reported by the authors. Table 3 represents the comparative study of newly discovered SARS-CoV-2, including other universal circumstances and other human viruses.

**Table 2.** Comparative study of newly discovered COVID-19 and four major human diseases [63].

| S. No. | Diseases | Symptoms | Onset of Disease | Incubation Period | Recovery | Complications | Transmission of Disease | Treatments | Remarks |
|---|---|---|---|---|---|---|---|---|---|
| 1. | Novel Coronavirus (COVID-19) | Fever, Cough Shortness of breath Fatigue | Sudden | 2–14 days after exposure | 2–8 weeks | Acute pneumonia, Septic shock, Respiratory failure | Human to Human | No vaccines available, only symptoms can be treated | This is a bulk infectious agent, respiratory syndrome virus |
| 2. | Severe Acute Respiratory Syndrome (SARS) | Fever Dry Cough, Headache Difficulty in breathing, Muscle aches, Loss of appetite Diarrhea | Sudden | 2–7 days after exposure | 5–6 weeks | Heart, Liver and Respiratory failure in adverse condition | Human to Human | Breathing ventilator to deliver oxygen. Pneumonia-treating antibiotics, Antiviral, medicines, Steroids to reduce lung swelling | The complete 29,727-nucleotide sequence of the RNA genome, deadly virus |
| 3. | Middle East Respiratory syndrome (MERS) | Fever, Chills Diarrhea Nausea Vomiting Congestion Sneezing, Sore throat | Sudden | 5–6 days after exposure | 6–7 weeks | Acute Pneumonia, Kidney failure in adverse condition | Human to Human | Treatment only for symptoms such as Fluids replacement and Oxygen therapy | causes devastating loss to human life, |
| 4. | Common Flu | Runny or Stuffy nose, Sneezing, Sore throat, Mild, Headache, fever | Gradual | 2–3 days after exposure | 7–10 weeks | Extremely rare or none | Human to Human | Symptoms can be treated by medication | This is a very small infectious agent |

**Table 3.** Comparison of newly discovered SARS-CoV-2 and other human viruses.

| S.No. | Virus Name | Symptom | Year | Countries Affected | Cases | Deaths | Fatality Rate (%) | Vaccine/Treatment |
|---|---|---|---|---|---|---|---|---|
| 1. | H7N9 Bird Flu | Difficult breathing, Fever, Cough, Runny nose | 2013 | 03 | 1568 | 616 | 39.30 | Symptoms can be treated |
| 2. | MERS | Fever, Cough, Cold, Difficult breathing | 2012 | 28 | 2496 | 858 | 34.40 | Symptoms can be treated |
| 3. | H1N1 | Fever, Chills, Cough, Sore throat, Body aches, Diarrhea, Vomiting | 2009 | 214 | >762,630,300 | 284,500 | 0.02 | PAnvax |
| 4. | SARS | Cough, Cold, Fever, Difficult breathing | 2002 | 29 | 8096 | 774 | 9.60 | Symptoms can be treated |
| 5. | Nipah | Fever, Headache, Myalgia, Pneumonia, Vomiting, etc. | 1998 | 02 | 513 | 398 | 77.60 | ChAdOX1NiV$_B$ |
| 6. | H5N1 Bird Flu | Difficult breathing, Fever, Cough, Runny nose | 1997 | 18 | 861 | 455 | 52.80 | Audenz |
| 7. | Hendra | Weakness, Fatigue, Stomach pain | 1994 | 01 | 07 | 04 | 57.00 | Equivac HeV |
| 8. | Ebola | Fever, Muscle and joint pain, Stomach pain | 1976 | 09 | 33,577 | 13,562 | 40.40 | Rvsv-zebdv |
| 9. | Marberg | Nausea, Vomiting, Chest pain, Sore throat | 1967 | 11 | 466 | 373 | 80.00 | cAds-Marburg |
| 10. | SARS-CoV-2 | Cough, Cold, Fever, Difficult breathing | 2020 | 222 | 174,054,314 | 3,744,116 | 3.4% | Vaccines available, symptoms can be treated |

## 7. Future Perspective and Management of SARS-CoV-2 Pandemic

The COVID-19 outbreak is proving to be an unprecedented disaster, especially in the most affected countries, including USA, India, Italy, Iran and China in all aspects, especially health, social and economic [110]. This could be mitigated by the effect of UV light on the survival of the virus on surfaces, climate-specific cultural differences, such as living more outdoors than indoors, immunological differences of the population, pre-exposure to coronaviruses or higher temperatures [110]. In addition, to this hopeful lower impact, if prevention measures are implemented, we could register a lower incidence of hygiene-linked diseases that still represent leading causes of death.

According to the scientist and researcher, a third wave of SARS-CoV-2 is imminent and will have a fatal effect on young children because of the multiplication of the virus [84]. This is why a very effective method and management medical system must be adopted to prevent its effect. Several investigational agents are being explored for antiviral treatment of COVID-19, and enrolment in clinical trials should be discussed with patients or their proxies. The following future perspective and management are needed during this current situation:

- During the pandemic, there is a bombardment of information about what to do and what not to do from different sources. The consequences of this must be considered.
- News media articles and social media posts have a tendency to sensationalize the outbreak and spread misinformation, creating panic. This must be addressed.
- At present, it is important to consider these factors to understand the experiences of people affected by COVID-19 and to make public health policy. Only by doing this will their mental health concerns to be addressed.

## 8. Conclusions

This review comprehensively discusses the SARS-CoV-2 global pandemic and its attributes, evaluation, detection methods, and problem and challenges along with treatment blueprints. The occurrence of COVID-19 has emphasized the importance of improving the mutually affective relationship between humans and the global environment. Environmental change is one of the most vital challenges of the 21$^{st}$ century and the coronavirus crisis has been declared a Public Health Emergency of International Concern by WHO. The SARS-CoV-2 responsible for COVID-19 received global attention because of its capability of causing community and healthcare-associated outbreaks of severe infections in human populations [93]. The present review concludes that SARS-CoV-2 spread more rapidly thanSARS, MERS, and other human viruses; the reasonsfor this areincreased globalization and the adaptation of the virus in every environment. At this point in time, it is imperative thatthe source of disease be controlled and its transmission pathcut off. The use ofexisting drugs and means to control the progress of the disease proactively is also indispensable. For improved efficiency, soap as well as sanitizers containing a minimum of 60% alcohol are recommended to makes it a better alternative to prevent infection. Globally, COVID-19 cases continued to increase and reached 173,005,553 active cases with 3,727,605 fatalities (by 8 June 2021). At the time of writing, a total of 222 countries are affected by the first and second waves of the SARS-nCoV-2 pandemic. We found that America and India are the most affected countries in the world due to their higher population densities. Several theories exist regarding the elevated risk of adverse events for patients who developed COVID-19. Few clinically approved antiviral drugs and vaccines effective against COVID-19 were reported to date by different health agencies. Finally, one health approach could play an essential role in fighting COVID-19,and other similar diseases, in the future. On the basis of social and human safety, we suggest that fundamental research focus on the detection, contaminants, treatments and analyses of viruses that are pathogenic to humans and have biological characteristics. This review also illustrates the spread of SARS-CoV-2 worldwide and concludes that it needs to be properly addressed and a vaccine developed so as to build a better future.

**Supplementary Materials:** The following supporting information can be downloaded at: https://www.mdpi.com/article/10.3390/covid2060055/s1, Table S1. The sequences of siRNA used in this study; Table S2. The primer sequences for qRT-PCR.

**Author Contributions:** All analyses, interpretations, and conclusions were planned and discussed jointly by both authors. The first author (R.K.) wrote the first draft of the ms and critically re-viewed the manuscript for important intellectual content. The second author (A.S.) collected the information, wrote and edited the entire manuscript. Both authors have read and agreed to the published version of the manuscript.

**Funding:** This research received no external funding.

**Institutional Review Board Statement:** Not applicable.

**Informed Consent Statement:** Not applicable.

**Data Availability Statement:** Not applicable.

**Acknowledgments:** Authors are thankful to Pt. Ravishankar Shukla University, Raipur, C.G., India for providing lab facilities and research platforms.

**Conflicts of Interest:** The authors declare that they have no known competing financial interest that could have appeared to influence the work reported in this paper.

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
