# Peer review of "An Overview of SARS-CoV-2 and Technologies for Detection and Ongoing Treatments: A Human Safety Initiative"

_covid, doi:10.3390/covid2060055_

Round 1
Reviewer 1 Report
May be accepted
Author Response
Responses to the Comments of Reviewer:
Reviewer #1
Title of Manuscript: An overview on SARS-CoV-2 and technologies for their detection and preliminary treatments: a human safety initiative
- May be accepted
Response: We (Dr. Ramsingh Kurrey and Anushree Saha) are very much thankful to respected reviewer for accepting our manuscript.

Reviewer 2 Report
It is an interesting review on noteworthy features of the world intimidating SARS-CoV-2 global pandemic along with its evaluation, problems and challenges in the 19 treatment strategies Comment1. The structure of the article with the different sections is very interesting, and the bibliographical support is very outstanding However, the part dedicated to the treatment could be highlighted a little more. In the title of the article, perhaps "preliminary treatments" could be replaced by "ongoing treatments" since there are different drugs with proven efficacy and admitted by the WHO and by all scientific societies 2. In the "treatment strategies" section, new drugs such as baricitinib, authorized by the WHO, and the human recombinant monoclonal antibodies casivirimab/imdevibad, have not been included. Nor is Sotrovimab, the humanized IgG1 monoclonal antibody, mentioned, nor is the new antiviral molnupiravir pending final evaluation by the VHO and the FDA. 3. Regarding vaccines, the following comment (line 417) is not accurate: "a vaccine needs to pass three clinical trials in order to go from laboratory to the clinic". During the COVID pandemic, the procedures have been extraordinarily shortened and mixed phase II and III studies have been carried out together, and permission has also been obtained for application to patients with their informed consent on the limited information available. In addition, it would also be interesting to comment on the extension of vaccination to adolescents and children over 6 years of age, authorized by the WHO, FDA and EMA. In table 3, page 20, the phrase “ No specific vaccines available, only symptoms can be treated” should be replaced 4. Regarding diagnostic procedures (line 502), it would be reassuring to comment that the new computed tomography (CT) devices emit little ionizing radiation, little more than 2 chest X-rays, and therefore it is a very safe procedure.
Author Response
Responses to the Comments of Reviewer:
Reviewer #2
It is an interesting review on noteworthy features of the world intimidating SARS-CoV-2 global pandemic along with its evaluation, problems and challenges in the 19 treatment strategies.
Response: We are very thankful for the valuable scientific remarks/comments made by esteemed reviewers to improve the quality of the submitted work. The suggestions of referee comments are fully responded point by point and highlighted in red color in the manuscript and blue color in the response letter. Hope, we satisfied the reviewers’ queries and suggestions in the revised version of this manuscript to meet the journal publication requirements. The responses to reviewer comments are given below.
Comments: 1. The structure of the article with the different sections is very interesting, and the bibliographical support is very outstanding However, the part dedicated to the treatment could be highlighted a little more. In the title of the article, perhaps "preliminary treatments" could be replaced by "ongoing treatments" since there are different drugs with proven efficacy and admitted by the WHO and by all scientific societies.
Response: Thank you very much for valuable scientific remarks by respected reviewer to improve the quality of the submitted manuscript. As per suggestion, this has been incorporated in the RMS. Please see the revised manuscript.
Title:
“An overview on SARS-CoV-2 and technologies for their detection and ongoing treatments: a human safety initiative”
- In the "treatment strategies" section, new drugs such as baricitinib, authorized by the WHO, and the human recombinant monoclonal antibodies casivirimab/imdevibad, have not been included. Nor is Sotrovimab, the humanized IgG1 monoclonal antibody, mentioned, nor is the new antiviral molnupiravir pending final evaluation by the VHO and the FDA.
Response: Thank you very much to reviewer for suggestion. As per suggestion, new drugs such as baricitinib, casivirimab/imdevibad etc. has been incorporated in the “treatment strategies” section in revised manuscript. Please see the Page No. 14 in RMS.
- Regarding vaccines, the following comment (line 417) is not accurate: "a vaccine needs to pass three clinical trials in order to go from laboratory to the clinic". During the COVID pandemic, the procedures have been extraordinarily shortened and mixed phase II and III studies have been carried out together, and permission has also been obtained for application to patients with their informed consent on the limited information available. In addition, it would also be interesting to comment on the extension of vaccination to adolescents and children over 6 years of age, authorized by the WHO, FDA and EMA. In table 3, page 20, the phrase “No specific vaccines available, only symptoms can be treated” should be replaced.
Response: We appreciate the reviewer comments; the correction has been made as suggested. Please see Page No. 15 and 21 and Table 3 in the RMS.
Table 3. Comparison of newly discover SARS-CoV-2 and other human viruses |
||||||||
S.No. |
Virus name |
Symptom |
Year |
Countries affected |
Cases |
Deaths |
Fatality rate (%) |
Vaccine/Treatment |
1. |
H7N9 Bird Flu |
Difficult breathing, fever, cough, runny nose |
2013 |
03 |
1,568 |
616 |
39.30 |
Symptoms can be treated |
2. |
MERS |
Fever, Cough, cold, difficult breathing |
2012 |
28 |
2,496 |
858 |
34.40 |
Symptoms can be treated |
3. |
H1N1 |
Fever, chills, cough, sore throat, body aches, diarrhea, vomiting |
2009 |
214 |
>762,630,300 |
284,500 |
0.02 |
PAnvax |
4. |
SARS |
Cough, cold, fever, difficult breathing |
2002 |
29 |
8,096 |
774 |
9.60 |
Symptoms can be treated |
5. |
Nipah |
Fever, headache, myalgia, Pneumonia, vomiting, etc. |
1998 |
02 |
513 |
398 |
77.60 |
ChAdOX1NiVB |
6. |
H5N1 Bird Flu |
Difficult breathing, fever, cough, runny nose |
1997 |
18 |
861 |
455 |
52.80 |
Audenz |
7. |
Hendra |
Weakness, fatigue, stomach pain |
1994 |
01 |
07 |
04 |
57.00 |
Equivac HeV |
8. |
Ebola |
Fever, muscle and joint pain, stomach pain |
1976 |
09 |
33,577 |
13,562 |
40.40 |
Rvsv-zebdv |
9. |
Marberg |
Nausea, vomiting, chest pain, sore throat |
1967 |
11 |
466 |
373 |
80.00 |
cAds-Marburg |
10. |
SARS-CoV-2 |
Cough, cold, fever, difficult breathing |
2020 |
222 |
174,054,314 |
3,744,116 |
3.4% |
Vaccines available, symptoms can be treated |
- Regarding diagnostic procedures (line 502), it would be reassuring to comment that the new computed tomography (CT) devices emit little ionizing radiation, little more than 2 chest X-rays, and therefore it is a very safe procedure.
Response: As per suggestion, this has been incorporated in the RMS. Please see the revised manuscript (Page no. 17).

Reviewer 3 Report
The data provided in the manuscript is almost 9-10 months old (Lines-12, 13 and 121-122, fig 2).
Some statements are redundant (line 57).
Treatment discussed includes some of the drugs which failed in the clinical trials and are no more recommended (table 1).
No update on the oral approved therapy like molnupiravir, paxlovid, etc.
Similar reviews have been published and it does not add anything to the existing knowledge/information.
Critical review or as the author mentioned in lines 104-105, as the systematic review is better than just compilation of the data.
Author Response
Responses to the Comments of Reviewer:
Reviewer #3
Response: We are very thankful for the valuable scientific remarks/comments made by esteemed reviewers to improve the quality of the submitted work. The suggestions of referee comments are fully responded point by point and highlighted in red color in the manuscript and blue color in the response letter. Hope, we satisfied the reviewers’ queries and suggestions in the revised version of this manuscript to meet the journal publication requirements. The responses to reviewer comments are given below.
- The data provided in the manuscript is almost 9-10 months old (Lines-12, 13 and 121-122, fig 2).
Response: Thank you very much for valuable scientific remarks by respected reviewer to improve the quality of the submitted manuscript. As per suggestion, this has been incorporated in the revised manuscript (RMS). Please see the RMS.
- Some statements are redundant (line 57).
Response: The redundant sentences are removed from revised manuscript. Please see the RMS.
- Treatment discussed includes some of the drugs which failed in the clinical trials and are no more recommended (table 1).
Response: The some of the drugs, which failed in the clinical trials, are separately described with scientific reasons in Table 1. Please see the Table 1 in the RMS.
- No update on the oral approved therapy like molnupiravir, paxlovid, etc.
Response: The oral approved therapy such as molnupiravir and paxlovid have been incorporated in the revised manuscript. Please see the Page No. 14 in the RMS.
- Similar reviews have been published and it does not add anything to the existing knowledge/information.
Response: This review aims to summarize the noteworthy features of the world intimidating SARS-CoV-2 global pandemic along with its evaluation, problems and challenges in the treatment strategies, clinical efficiency and detection methods are proposed so far.
- Critical review or as the author mentioned in lines 104-105, as the systematic review is better than just compilation of the data.
Response: Thank you very much for valuable scientific remarks.

Reviewer 4 Report
The review authored by Ramsingh Kurrey and Anushree Saha has done a very elaborate discussion on the SARS-CoV-2 global pandemic and has highlighted the current technologies that are being used to detect SARS-CoV-2 infection. The review has also done a good job of describing the burdens, challenges, treatment options, and potential management regimens that could be implemented in India and similar countries.
Overall, the review is well written and has an extensive summary of SARS-CoV-2 infection and challenges in the field!
Author Response
Responses to the Comments of Reviewer:
Reviewer #4
The review authored by Ramsingh Kurrey and Anushree Saha has done a very elaborate discussion on the SARS-CoV-2 global pandemic and has highlighted the current technologies that are being used to detect SARS-CoV-2 infection. The review has also done a good job of describing the burdens, challenges, treatment options, and potential management regimens that could be implemented in India and similar countries.
Overall, the review is well written and has an extensive summary of SARS-CoV-2 infection and challenges in the field!
Response: We both are very much grateful for the valuable scientific remarks/comments made by esteemed reviewer to appreciate the submitted work. Thank you so much.

Round 2
Reviewer 3 Report
The manuscript has been thoroughly revised. Changes made are acceptable to me.